# Sexuality (and Lack Thereof) in Adolescence and Early Adulthood: A Review of the Literature

**DOI:** 10.3390/bs6010008

**Published:** 2016-03-17

**Authors:** Marie-Aude Boislard, Daphne van de Bongardt, Martin Blais

**Affiliations:** 1Département de Sexologie, Université du Québec à Montréal, Case postale 8888, Succursale Centre-Ville, Montréal, QC H3C 3P8, Canada; blais.martin@uqam.ca; 2Research Institute of Child Development and Education (YIELD), University of Amsterdam, P.O. Box 15776, Amsterdam 1001 NG, The Netherlands; D.vandeBongardt@uva.nl

**Keywords:** adolescence, early adulthood, sexuality, sexual behavior, development, normative, sexual minorities, casual sexual relationships and experiences (CSREs), literature review

## Abstract

Youth sexuality has been primarily studied with a focus on its potential public health issues, such as sexually transmitted infections and unwanted pregnancies, and its comorbidity with other risky behaviors. More recently, it has been studied as a normative step in romantic partnerships, either pre- or post-marital, as well as outside the context of romantic involvement. In this paper, we review the extensive literature on sexuality in adolescence and early adulthood both *within* and *outside* romantic relationships (*i.e.*, casual sexual relationships and experiences; CSREs). Furthermore, the recent recognition of youth sexuality as a developmental task has led to a renewed interest from scholars in youth *who abstain from sexual encounters*, whether deliberately or not. A brief overview of the literature on cultural differences in sexuality, and sexual-minority youth sexual development is also provided. This paper concludes by suggesting future directions to bring the field of youth sexuality and romantic relationships forward.

## 1. Introduction

Research on the psychosexual development of adolescents is generally characterized by two main streams. First, adolescent (or premarital) sexual activity has been investigated as potentially risky and harmful, and examined from a public health perspective for a long time [1,2,3,4]. This traditional line of research is of critical importance, considering the high prevalence rates of sexually transmitted infections (STIs) and unwanted teenage pregnancies in countries such as Great Britain and the United States [5,6,7]. Youth risky sexual behaviors, commonly operationalized as precocious sexual onset (*i.e.*, first intercourse before 15 years old; [8,9]), inconsistent condom use, and multiple sexual partners [10], are directly involved in the high prevalence rates of STIs, including HIV [11]. Past decades of research have yielded important knowledge on the improvement of prevention and intervention strategies in education and health care aiming to promote youth sexual health [12,13,14].

Second, although research on adolescent risky sexual behavior is of vital importance, over the past two decades, it has been increasingly recognized that the exploration of intimate relationships and sexual behaviors during adolescence and emerging adulthood are not inherently risky. As such, the field of youth sexual development has recently shifted toward an increased recognition of sexual intimacy with one’s partner as a developmental task in adolescence and emerging adulthood [15,16,17]. In this expanding second line of research, scholars do not only focus on potentially risky aspects of young people’s sexual behavior but also on positive outcomes associated with sexual activity in adolescence and emerging adulthood. For instance, two studies found that female sexual subjectivity (defined as feelings of sexual self-efficacy, entitlement to sexual pleasure from self and partner, sexual body esteem and sexual self-reflection; [18], increases significantly with sexual experience [19,20]. Other recent longitudinal studies reported a more positive body image among male college students after their first experience of intercourse, and lower psychological distress among both male and female college students [21,22]. Another study observed that youth sexual health is associated with greater well-being in early and later adulthood [23].

This body of literature has provided insight into how comprehensive sexuality education can effectively promote responsible sexual decision-making in young people [24,25,26]. Thus, youth sexuality research agenda is now taking both its normative and risky components into account, focusing on promoting positive sexual health [20,27,28] and prevention of sexual health issues such as STIs and HIV, unplanned pregnancies, sexual coercion and abuse, and violence in romantic relationships. This is also in line with the World Health Organization’s [29] claim that sexual health is not merely the absence of illness or sexual problems, but also encompasses physical, mental, emotional, and social well-being in relation to sexuality.

## 2. Timing of Sexual Onset

Research has shown that most heterosexual adolescents follow a *progressive sexual trajectory*, in which they first engage in non-genital behaviors (e.g., kissing, holding hands, hugging), followed by genital sexual behaviors, (e.g., mutual touching of the genitals, oral sex), and, ultimately, vaginal intercourse [30,31]. The majority of adolescents in the Netherlands, Canada, and in most Western countries follow this developmental sequence [31,32], although there are individual differences in the pace of this sequence. As such, sexual repertoires typically increase in diversity along with age [33], but apart from a few studies on oral sex (e.g., [34,35]), the vast majority of the literature on adolescent sexual behavior has focused on intercourse.

Research shows that a majority of teenagers from the U.S. [36,37], the UK [38], Canada [39]), Australia [40] and the Netherlands [41] report experience with sexual intercourse by eighteen years of age. Between 10% to 40% of youth in the U.S. and other Western countries remain virgins after age 18 [42,43]. These rates drop dramatically among older emerging adults, with only 5% of males and 3% of females still being virgins at ages 25 to 29 [44]. Hence, individual differences in *timing of first intercourse* have been one of the most studied features of sexual development [45]. A lot of attention has been devoted to the factors associated with first intercourse in adolescence, as well as to targeting the protective factors to delay its onset. As a result, numerous studies have been published on the individual, interpersonal, and socioeconomic characteristics that precipitate or delay sexual initiation, and there is abundant literature on the correlates of the distinct trajectories of early, normative, and late first intercourse (e.g., [45,46,47]. Recent longitudinal evidence suggests that both early *and* late sexual onset are related to mental and sexual health issues, poorer peer relationships, and higher adjustment difficulties in comparison with a more normative timing of sexual debut [4,8,15,45,48]. These findings echo with *lifespan* [49] and *lifecourse* [50] theories that raise the importance of transitioning *in sync* with peers within a specific cohort, historical period, cultural context, and within group norms. These theories suggest that *early* and *late* normative life transitions are associated with increased challenges, over and beyond the difficulties of any transition *per se*.

*Early Sexual Onset*. While early engagement in sexual behaviors is not *inherently* problematic [51], research indicates that early starters are typically more vulnerable to potential risks. It has been argued that teenagers in early and middle adolescence are generally not “cognitively ready” for safe and consensual sexual interactions [52]. Moreover, younger adolescents are generally more impulsive [53] and more sensitive to social pressure [54]. They also often have less knowledge about sexual risks, and tend to be less confident and assertive during interactions with partners [41]. As a result, early initiators are more likely to have condomless sex [9,55] , to accumulate more sexual partners throughout adolescence [9], to have non-consensual sexual experiences [41], to contract sexually transmitted infections [56], to become pregnant as a teenager [7], and to experiment subsequent increases in externalizing behaviors, albeit only among females [4]. Furthermore, sexual precocity is often comorbid with both externalized symptoms [8], especially among boys [57] and internalized problems [58], such as low self-esteem among girls [59]. In light of such potential risks, research on factors that delay or promote the initiation of sex is paramount. Yet, the scholars’ attention has been devoted to the correlates of *early* sexual intercourse in adolescence, neglecting the individuals who remain sexually abstinent into adulthood.

*Late Sexual Onset***.** Until recently, literature on sexual abstinence in late adolescence and emerging adulthood was scarce and tended to present virginity as a personal choice [60], based on religious attendance and religiosity [61], moral principles and conservative attitudes [62], or high academic goals. Other individual characteristics identified as correlates of virginity in late adolescence and early adulthood include a younger appearance due to late pubertal development [63], and lower use of alcohol and drugs [64]. When adolescent abstainers were asked directly why they refrained from sexual activity, various reasons emerged from their responses: fear- or uncertainty-based postponement (e.g., not ready, fear of pregnancy), conservative values (e.g., religious values, desire of waiting until marriage), and emotionality and confusion [65]. Similar findings from a Dutch study on sexual attitudes, behaviors, and health of adolescents and young adults aged 12–25 years [41] indicated developmental patterns in the reasons for not having sex. Whereas most 12–14 years old (72%) mentioned considering themselves too young, this percentage rapidly decreased to 34% among 15–17 years old, and 11% among 18–24 years old. Among 15–24-year olds, the most cited reason was that it just had not happened yet (49%–56%). However, various other reasons were also cited (e.g., being scared, not finding an eligible sexual partner, and parental or religious constraints), showing individual variation in reasons for abstaining from sex, and highlighting the need to increase our understanding of complex psychosocial processes behind young people’s sexual decision making.

In addition to the aforementioned individual factors, several studies have also identified interpersonal factors associated with adolescent virginity, mostly related to family structure and education and peer influences. For family influences, living with both biological parents [63], having a highly educated mother [64], and believing that adults and parents have high achievement standards toward them are linked to delayed sexual onset in youth [66]. With regards to peer relationships, numerous studies have shown that adolescents who postpone their first intercourse until later in life are more likely to have friends who also believe in delaying sexual intercourse [45,67] and who are involved in religious activities [68]. These findings are in line with the extensive research of the last 40 years supporting homophilia in friend selection among adolescents [69,70,71], including among late sexual starters who tend to hang out with alike-peers [47] in groups where sexual abstinence is the norm [63].

*Virginity in Early Adulthood*. Most of the literature on sexual delay and virginity in adolescence has documented their “positive” correlates. However, when virginity continues into early adulthood, different factors are likely to explain this delay. Since the paradigm on adolescent sexuality has shifted from a risky vision leading to abstinence-only sex education to the recognition of sexuality as a developmental task of the second decade of life, an increasing number of studies are now documenting the correlates and predictors of virginity in early adulthood. For instance, one study showed that adult virgins have higher odds of being overweight and of being perceived as physically unattractive [72]. Four additional studies reported that adult virgins have greater probabilities of never having been in a romantic relationship [47,68,73,74]. Moreover, in a qualitative study conducted among 82 involuntary celibate adults aged 18 to 64 years, Donnelly and colleagues [48] found that nearly all adult virgins never dated anyone, including in adolescence. Thus, findings converge to support the importance of romantic and sexual experiences during adolescence for ongoing romantic and sexual development in adulthood. Additionally, this study revealed that these adult virgins perceived themselves as being very shy and unable to establish social contacts, and reported body image issues, such as being overweight and perceiving their physical appearance to be an obstacle to their sexuality [48]. In summary, research on adulthood virginity, although embryonic, has started to uncover that there are several reasons for remaining a virgin in early adulthood, some relating to personal choices, and others being more egodystonic and linked to a lack of sexual opportunities.

Therefore, one of the issues for scholars focusing on virginity is the diversity among this population, and the distinction between youth who have never had dyadic erotic experiences, on the one hand, and those who abstain from coitus but engage in other partnered sexual behaviors, on the other. The latter are often referred to as “technical virgins” in the literature [75]. These individuals have partnered sexual repertoires that include various sexual experiences, like mutual touching of the genitals, oral sex, and more rarely anal sex, but refrain from engaging in penile-vaginal intercourse [75,76]. Various reasons can motivate this technical virginity, ranging from religiosity to wanting to avoid potential negative consequences of sexual activity [34], or waiting for the “right person” [75]. Sexual-minority youth who have never engaged in sexual intercourse with someone of the other-sex are also often confounded with heterosexual virgins if no questions on sexual orientation are asked [47]. Thus, technical virgins seem to be in different psychosexual developmental trajectories than those who never experienced any sexual contact with a partner because of difficulties to attract one, or of very little interest in sexual interactions [77]. Because most studies have conflated these two distinct populations, our knowledge on the prevalence and characteristics of emerging adults without any sexual experience with another person remains very limited [72].

*Asexuality.* One of the explanations associated with sexual abstinence in youth is asexuality. According to the rare surveys in which such data is available, up to 1% of the general population is asexual [77], that is, individuals “*who, regardless of physical or emotional condition, actual sexual history, and marital status or ideological orientation, seem to prefer not to engage in sexual activity*” ([78], p. 97). The question of whether asexuality constitutes a sexual orientation in itself has been an ongoing a debate among sex scholars [77,78,79]. Even though asexual youth are more likely than sexual ones to be virgins in adulthood [77], the overlap between adulthood inexperience and sexual non-attraction is incomplete. For instance, in Haydon and colleagues’ study [72], only half of sexually inexperienced adults of both genders reported never having been sexually attracted to someone. However, the links between sexual attraction, sexual experiences, and sexual identity are complex and still largely overlooked [80,81].

## 3. Gender Differences in Youth Sexuality

Although historical changes have taken place in gender expectations regarding sexuality, recent studies suggest that the sexual double standard persists. Stricter social norms remain for female sexuality, encouraging young girls to refrain from sex and avoid cumulating multiple sexual partners [82], whereas boys are generally granted more sexual freedom. Girls are also more often discouraged by their peers from having sex [83,84,85], whereas boys are generally socialized in more sex-positive peer contexts, characterized by more approval of—but also more pressure toward—sexual activity, especially from male friends [86]. Accordingly, boys generally report higher rates of masturbation and lifetime sexual partners and more consistent sexual activity than girls in studies on youth sexuality [87].

Even more striking are the reported differences in emotions associated with sexual activity across genders. Qualitative research has found that many young girls experience ambivalence and mitigated emotions following their experience of first intercourse [88,89]. This has recently been substantiated by a quantitative study conducted among Dutch adolescents, in which sexually experienced boys reported overall more positive sexual emotions than sexually experienced girls [90]. More specifically, boys reported experiencing more pride after sex, whereas girls were more likely to report feeling “dirty” and shameful [90]. Similar findings have been reported in another quantitative study by Brady and Halpern-Fisher [82]: sexually inexperienced girls aged 14 to 16 years old reported more mixed feelings about their virginity than boys. The authors hypothesized that young girls may feel pressured by their romantic partners to have sex, while feeling pressured by most other socialization agents, including parents and same-sex peers, to abstain from sex. This interesting avenue has yet to be empirically tested.

This sexual double standard was also observed in Carpenter’s qualitative interview study [91], in which more women perceived their virginity as a *gift* (*i.e.*, to be given to a beloved partner), whereas more men viewed theirs as a *stigma* (*i.e.*, to be eradicated as soon as possible). This finding was also replicated in a quantitative study presenting the same virginity scripts to a different emerging adult sample [92]. As a result of viewing virginity as a stigma, males are more likely to perceive it as a source of embarrassment [91], and to lie about their virgin status [93]. Furthermore, a study conducted among virgin adolescents found a higher proportion of males reporting being a virgin as a lack of sexual opportunities when compared to females [94]. There is also evidence suggesting that males who delay sexual onset until adulthood are more likely to have anxious symptoms than on-time males [45].

In summary, the evidence converges in that sexual scripts, sexual trajectories, cognitions about sexuality, and the subjective experiences of dyadic sexuality differ by gender. However, meta-analyses investigating gender differences in sexual attitudes and behaviors have also found substantial *similarities* between males and females [87,95]. This supports the *gender similarity hypothesis*, which hold that, overall, boys/men and girls/women tend to be more psychosocially (and sexually) similar than different [96]. This suggests that within-gender individual differences in sexuality may be stronger than between-gender differences, and points to an important site for future research.

Furthermore, an interesting study from Tolman and Diamond [97] pointed out that research on male and female sexuality over the life course lacks theorization, resulting in a fragmented knowledge from studies focusing either on sociocultural and political aspects leading to gender differences in sexual development, or on the biological aspects of gendered sexual behavior. In addition, a multimethod study assessing gender differences in values among 127 samples issued from 70 countries revealed that gender differences are small (median d = 0.15) and typically explain less variance than age and much less than culture [98]. Together, these studies suggest that gender differences observed in youth sexuality may be better explained by cultural factors and socialization effects than by the biology distinguishing males and females. Since sexual norms are shaped by the more distal influences of culture, gender differences in youth sexual development may be a reflection of a gendered socialization in any particular culture.

## 4. Cultural Differences in Youth Sexuality

With regards to youth sexual development and sexuality across cultures, research has revealed both commonalities and cultural specificities in different subgroups based on their cultural background and context. A systematic review of 268 qualitative studies on young people’s sexual behaviour published between 1990 and 2004, including studies on cultural minorities and among culturally diverse samples, explored common themes in young people’s sexual lives across cultures. This review pointed out seven key themes that were not exclusive to any particular country or cultural background, specifically: (1) young people assess potential sexual partners as “clean” or “unclean”; (2) sexual partners have an important influence on behaviour in general; (3) condoms are stigmatising and associated with lack of trust; (4) gender stereotypes are crucial in determining social expectations and, in turn, behaviour; (5) there are penalties and rewards for sex from society; (6) reputations and social displays of sexual activity or inactivity are important; and (7) social expectations hamper communication about sex [99]. These seven themes were present, in varying degrees, in all countries assessed. This study suggests that some attitudes toward sexuality might be universal in youth. Despite these commonalities, many studies have examined differences in the sexual behaviors, attitudes and development of youth with specific cultural backgrounds, and have done so in two different ways: (1) by comparing collectivist *versus* individualistic cultures, and assessing different dimensions of youth sexuality while conducting sex research in non-Western countries; and (2) by comparing youth sexuality with samples from diverse cultural backgrounds in countries marked by high rates of immigration. The main findings from each of these two lines of research are reviewed below.

*Youth Sexuality in Collectivist* versus *Individualistic Cultures.* Based on one of Hofstede’s cultural dimensions for cultures [100], scholars have investigated whether youth raised in collectivist cultures (*i.e.*, cultures that place great value in social belonging and group responsibility; e.g., some African and Asian countries) differ in their sexual development from those who grow up in more individualistic countries (*i.e.*, cultures that emphasize the value of independence and individual well-being; e.g., the United States, Canada, Australia, and Western Europe). In general, the former tend to be more oriented toward their social context, show more sensitivity and conformity to social norms, and greater endorsement of friendship rules than the latter [101,102,103,104]. In terms of sexuality, they are also typically surrounded by sexual norms that tend to be overall more conservative in comparison with those in more individualistic cultures [105]. It has been suggested that, as a result of these cultural characteristics, youth raised in collectivist cultures are more susceptible to social influences in the development and shaping of their sexuality and sexual decision-making (e.g., [90]). Indeed, a meta-analysis on peer influences on adolescent sexual activity conducted by Van de Bongardt, Reitz, Sandfort, and colleagues [90] found that friends’ sexual behaviors and peer pressure to have sex were more strongly related to the levels of sexual activity of adolescents in collectivist cultures than in individualistic cultures. However, this meta-analysis also revealed that adolescents’ perceptions of their peers’ sexual activity had the strongest effect on their own sexual activity, compared to other types of sexual peer norms (*i.e.*, peer sexual attitudes, peer pressure), regardless of the country in which the included studies were conducted. Thus, the fact that perceptions of peer sexual behavior are an important proximal factor in adolescents’ sexual decision-making across countries and cultures again shows that not only cross-cultural differences but also similarities can be found.

*Sexual Development of Ethnic Minority Youth*. A considerable amount of research has also been devoted to determining the effects of ethnic group membership on youth sexual development. In a review of 35 longitudinal studies on age at first intercourse, mostly conducted in the U.S., Zimmer-Gembeck and Helfand [45] reported that, after controlling for socioeconomic status and parental education, the 13 studies that included ethnicity as a predictor showed earlier onset of sexual intercourse for Black males, but not Black females, when compared to White adolescents. Hispanic adolescents reported an age of first intercourse similar to White adolescents, and Asian American adolescents reported a later onset of sexual activity. Findings from the representative large-scale The Add Health survey in the U.S. also revealed that virgin males had lower odds of initiating sexual activity after age 18 if they were non-Hispanic Asian [72]. Furthermore, although the evidence usually shows that Black and White girls do not differ in their average age of first intercourse, or in their rates of early onset, regional differences may exist. For example, only a study from the southeast of the U.S. reported that Black females had their first experiences of sexual intercourse earlier than White females [106]. A meta-analysis from Wells and Twenge [107] examining differences in sexual behavior among young Whites, Blacks, and Latinos corroborated that Black adolescents often initiate intercourse at a significantly earlier age and that a higher percentage of Black adolescents are sexually active compared to Caucasian and Latinos, even after controlling for other sociodemographic factors.

Several studies have shown that adolescents from some minority ethnic groups in the United States (e.g., African American, Latino) are particularly at risk of negative sexual health outcomes, including early sexual initiation, unprotected intercourse, and high rates of STIs and teenage pregnancies [108,109,110]. This has been explained in terms of cultural beliefs and values regarding sexuality, socio-economic status, and social phenomena such as segregation, discrimination, and racism [108,109]. For instance, having experienced discrimination has been found to increase African American adolescents’ affiliations with deviant peers, which, in turn, promoted risky sexual behavior [111].

Differences between ethnic groups have also been identified in other countries. Yahyaoui and colleagues [112] examined the sexual attitudes and experiences of youth aged 13 to 20 living in France, who were either immigrants from countries in the Maghreb, a region highly influenced by Islam, or French-natives, mostly influenced by Christianity. They found that among the immigrant group, the sexual attitudes and sexual experiences were more traditional, more strongly affected by sexual taboos outside marriage and by fear of judgement. The authors also found that immigrant girls conformed to the sexual norms conveyed by their parents, culture and religion, and strongly endorsed the abstinence until marriage script, partly to avoid stigmatization and exposure to public slander. On the contrary, boys in the immigrant group reported having full access to sexuality, based on both their gender and the sociocultural traditions in their country of residence (*i.e.*, France). This result is consistent with Baumeisters’ examination of how historical and cultural changes affect male and female sexual behavior differently [113]. Through his theory of “female erotic plasticity” the author concluded that, although mostly based on cross-sectional research, female sexuality is generally more subject to sociocultural influences than male sexuality (*i.e.*, more socioculturally malleable), and that “variations among the societies in sexual customs are apparently greater for girls than for boys” (p. 325). Together, these studies suggest differential effects of culture on gender, though additional studies in other countries are needed to fully capture the possible interactions between culture and gender influences on youth sexuality.

Altogether, the literature on youth sexuality outside the Western world is limited, possibly due to a lower social recognition and funding of research on what can be considered a sensitive topic. As a result, the existing empirical knowledge lacks information on the lived experiences of about 90% of adolescents worldwide, who grow up in the “majority world”, which includes Asia, Africa, Latin America, and the Caribbean [114]. This is problematic, because the norms and values related to young people’s sexuality are likely to vary across these cultural contexts [115,116], leading to only partial understanding of the processes through which culture may shape youth sexuality.

## 5. Youth Sexual Behaviors in Romantic Relationship Contexts

Research from the U.S. shows that, in adolescence, most sexually active teenagers engage in (first-time) sexual behaviors within the context of a romantic relationship [117,118]. Moreover, findings from the Netherlands [41] indicate that many young people consider a romantic relationship context a normative prerequisite for having first-time sex. Specifically, when *non-sexually active* 12–25 years old were asked why they had not yet had sexual intercourse, 25% of the boys and 33% of the girls reported wanting to be in love first, and 34% of the boys and 47% of the girls mentioned first wanting to be in a dating relationship for a while. The motives for engaging in the first sexual intercourse experience further illustrated the importance of romance in the initiation of sex. Among *sexually active* youth, having been in a dating relationship for a while (53% of the boys and 67% of the girls), and being in love (60% of the boys and 73% of the girls) were mentioned as reasons for engaging in sexual intercourse for the first time. Furthermore, Maas and Lefkowitz [119] found that American University students who were either currently in a serious romantic relationship, or who had more romantic relationships in prior semesters, reported higher frequencies of kissing, oral sex and penetrative sex, indicating that romantic activity and sexual activity tend to go hand in hand.

Romantic relationships are not only normative contexts for young people’s emerging and ongoing sexual behaviors, but also appear to be contexts that promote more positive sexual experiences. Maas and Lefkowitz [119] found that romantically active university students (either currently or previously) reported more sexual esteem (*i.e.*, a higher evaluation of their sexual self, e.g., being a good sexual partner), suggesting that romantic relationships are a context to practice and improve sexual skills and confidence. Alternatively, the authors propose that is also possible that individuals with higher sexual esteem may be more successful in selecting, attracting, and maintaining potential romantic partners [119]. When looking at actual sexual satisfaction, another study among American university students found that students in exclusive dating relationships (e.g., cohabiting or engaged) reported more physiological/physical and psychological/emotional sexual satisfaction than students who were not dating or dating casually [120]. Similar findings were observed in a study among Norwegian college students, where sexually active yet romantically unattached young adults were less satisfied with their sex lives than those who were involved in long-term, committed relationships [121]. However, in this longitudinal study, a decrease in sexual satisfaction was observed over time, showing a negative link with relationship duration. The authors suggest that this may indicate that routine and boredom may negatively affect sexual satisfaction within long-term romantic relationships [121], but this hypothesis is yet to be investigated. Together, these findings illustrate the interplay between romantic experiences, sexual behaviors, and sexual skills, and the need to further investigate these associations, including using longitudinal designs.

*Sexual Experiences as Predictors of Romantic Relationship Quality.* When considering the interplay between romantic experiences and sexual behaviors, the bidirectionality of association needs to be taken into account. In other words, romantic relationship status or quality may not only affect sexual behaviors and the evaluation thereof, but the typology and quality of sexual behaviors engaged in may also affect the evaluation of the romantic relationship. In a study on adolescent heterosexual romantic couples aged 14–21 years who had been dating for at least four weeks, sexual experiences were indeed found to be predictive of the perceived quality of the romantic relationship [122]. Specifically, couples who engaged in more kissing were more satisfied with their relationship, and also more committed to their relationship [122]. The frequency of other sexual behaviors (*i.e.*, intimate touching, oral sex, and intercourse), however, were not overall predictive of relationship satisfaction or commitment, although younger adolescents who more frequently engaged in intercourse reported lower satisfaction [122]. Adolescents who felt more desire for their romantic partners (*i.e.*, experiencing romantic excitement, and interest in their partners’ body) also scored higher on relationship satisfaction and commitment [122]. This bidirectional interplay between romance and sexuality, the underlying mechanisms, and variation between couples as well as within couples (e.g., changes over time) needs to be further explored.

*Safe versus Risky Sex within Romantic Relationships.* Committed romantic relationships are generally considered safe contexts for sexual behaviors in comparison with casual sexual encounters. Research among American female adolescents shows that girls who had their first intercourse experience with their regular dating partner or their steady partner were more likely to use some form of contraceptives at first intercourse (*i.e.*, 75.2% and 76.4%, respectively), compared to girls who had first-time intercourse with a friend (56.1%) or someone they had just met (48.3%) [117]. However, it has also been argued that the perceived safety within romantic relationships may also cause young people to misjudge potential risks. In a qualitative study among 57 Australian adolescents, analyses of semi-structured interviews illustrated that many adolescents framed their sexual relationships as part of their search for love [123]. Furthermore, the interviews suggested that condom use negotiations with romantic partners were difficult to combine with notions of trust as a central component of a committed relationship [123]. As such, young people’s scripts for sex within romantic relationships seem to be an important site for the promotion of safe sex negotiation skills.

## 6. Youth Sexual Behaviors outside Romantic Relationship Contexts

Although most adolescents and young adults engage in sexual behaviors within the context of a romantic relationship, it is striking that in all the research on youth romance and sexuality, it is the field of sexual behaviors *outside* of committed romantic relationships, (*i.e.*, hook-ups, friends with benefits, one-night stands, and other types of sex partnerships) that has been expanding the most rapidly over the past decades, especially among emerging adults (see [124], for a review of literature). The expression *casual sexual relationships and experiences* (CSREs) is often referred to as the most inclusive way to describe these non-romantic sexual experiences [125]. There is a consensus among scholars that the majority of emerging adults, at least those attending colleges, participate in *hookups* [126]. In fact, lifetime prevalence data suggests that up to three quarters of college men and women have experienced CSREs [127,128]. Numerous studies reveal a spectrum of sexual experiences among adolescents and emerging adults that ranges from one-time sexual encounters to sexual relationships that take place only in a committed, romantic relationship. The term *hookup* is also often used to describe uncommitted sexual encounters, and encompasses other vernacular such as *one-night stand*, *booty call*, *friend with benefits*, *fuck buddy*, or *sexual encounters with no strings attached*. Despite increasing research on these non-romantic sexual experiences, few studies provide clear definitions or criteria to distinguish the various CSREs. However, as Claxton, DeLuca, and Van Dulmen [129] reported, most of this literature is based on the premise that CSREs are characterized by a lack of emotional connection and commitment between sexual partners. They are often considered unstable relationships, emotionally inconsequential interactions [130], or non-relational or non-relationship sex [131].

*CSREs Variety and Definitions.* While the main script of *hookups* implies the absence of a committed relationship, a non-relational and short-term encounter, and a variety of sexual behaviors [132], recent studies point to multiple scripts and a more nuanced portrait, especially regarding the first two components. Qualitative (e.g., [133]) as well as quantitative studies (e.g., [134]) have identified multiple criteria to describe CSREs, such as the relationship status when the partners first met (e.g., strangers, acquaintances, friends, et cetera) and their current relationship status, if applicable (e.g., acquaintance, dating partner, ex-romantic partner, friend, et cetera), the frequency of sexual contacts *versus* social activities, the importance attributed to sex as the primary goal of the relationship, the presence and explicitness of an agreement regarding sexual exclusivity, and the level of personal disclosure.

The combination of these criteria allows for several scripts that better reflect the diversity of sexual experiences and relationships that fall outside of a committed relationship. At least five different types of CSREs have been described. The *one-night stand* refers to “a sexual encounter with another individual that only occurred one time” ([135], p. 5). Because *one-night stands* occur not only with strangers but also with acquaintances and friends, Rodrigue and colleagues [134] suggested the label “one-time sexual encounter”. The *booty call* designates a non-committed, non-monogamous relationship between individuals, usually friends or acquaintances, who communicate with each other mainly in order to schedule sexual encounters [136]. Rodrigue and colleagues [134] refer to this profile as “mostly about sex partnership”. In the *one-time sexual encounter* and the *mostly about sex partnership* profiles, both social activities and personal disclosure are infrequent, and sex is the main goal of the encounter.

Another class of CRSEs describes relationships that combine aspects of both sexual relationships and friendships, or, in vernacular, *friends with benefits* relationships. Epstein and colleagues [132] reported that the central themes of the *friends with benefits* script are that the partner is a friend or an acquaintance, that a sexual activity is ongoing and that there is no monogamous commitment to each other. Rodrigue and colleagues [134] also report that such CRSEs come with a high frequency of both sexual and social activities, as well as high personal disclosure to one another. While they mostly report a non-monogamous sexual agreement, it is noteworthy that more than half of the respondents in intimate and sexual partnerships in Rodrigue and colleague’s [134] study had negotiated their sexual agreement explicitly. It is possible that some of those engaging in such intimate and sexual partnerships are transitioning to a couple relationship while rejecting monogamy. Rodrigue and colleagues [134] also report a *friendship first partnership* profile, which is a variation of *friends with benefits* in which sexuality is infrequent and not central to the friendship, characterized by high levels of personal disclosure and social activities.

Finally, sexual encounters outside of a committed relationship can also involve an ex-romantic partner. This type of CSRE involves a high level of personal disclosure, as well as frequent sexual and social activities. Described as a subtype of *friends with benefits* by Mongeau, Knight, Williams, Eden, and Shaw [137], this type of sexual relationship may capture the process of transitioning out of a couple relationship. However, there are distinctive characteristics between friends with benefits and romantic partners, as the former devoted more of the time spent together to sexual activity, practiced safe sex more frequently, communicated more often about extra-dyadic sexual experiences than the latter [138].

As these descriptions suggest, the majority of CSREs tend to occur with previously known partners, including in *one-night stands*, and often involve more than one encounter. Bisson and Levine’s results [139] suggest that CSREs come with various degrees of commitment, intimacy, and passion. Epstein and colleagues [132] also report that most men actually reject non-relational scripts of friends with benefits relationships, opting for a script that allows a greater relational connection. This portrait challenges the assumption of a lack of emotional connection and/or commitment between casual sexual partners that dominates the literature on CSREs, and questions the apparent distinction between friendships and romantic relationships.

*Sexual Scripts in CSREs.* A large body of research has described the changes in the sexual and relational scripts that took place during the twentieth century. Not only is sexual activity now integral to most dating and courtship scripts, but sex has also become accepted between partners who have no expectation of future contact or any intention of engaging in a committed, romantic relationship. Both genders still express at least some preference for dating over hooking up [140]. While Epstein and colleagues [132] propose that the presence of some kind of sexual behavior is a core definitional element of *hookups* and that it should minimally imply some level of nudity, there is little evidence that CSREs actually involve different sexual behaviors than romantic relationships. The variety of behaviors reported in both types of sexual relationships can include kissing, petting, breast and genital touching, masturbation and oral sex, and penetrative sex [125,141,142,143,144,145,146,147]. However, LaBrie, Hummer, Ghaidarov, Lac and Kenney [148] report that the types of sexual behaviors enacted vary according to the familiarity of the partners. For both males and females, hooking up with a familiar partner led to the furthest physical extent of penetrative sex as the most common response, as opposed to an unfamiliar sexual partner. With unfamiliar partners, however, males reported touching below the waist as the most common furthest extent, while it was kissing among females.

There is some evidence of an imbalance in the (hetero)sexual hookup script in favor of men [126]. Hookup is gendered in three ways according to England and colleagues [143]. There is an imbalance in the initiation script, where the man is expected and more likely to initiate sexual activities than women. Men also report having more orgasms and sexual satisfaction than women during a hookup, suggesting a gendered orgasm gap. Moreover, Backstrom, Armstrong, and Puentes [149] report a feminization of oral sex, where men are less likely than women to perform oral sex in hookups. Women also more often feel pressured by their male hookup partners to exceed their personal sexual boundaries [146]. Finally, there is a sexual double standard that stigmatizes both men and women, where women who have many sexual partners are labeled as “sluts” and men labeled as “man whores”. Both of these labels use words originally assigned to promiscuous women. However women still tend to be judged more negatively than men for participating in CSREs [150].

*Factors Associated with Involvement in CSREs.* At the behavioral level, prior hookups [144,148,151,152] and past alcohol use [129] tend to increase hookup behavior. At the personal level, involvement in CSREs is often described as a result of a compromised well-being, including depressive symptoms and suicidal ideation [153], and lower self-esteem [144]. Traits like impulsivity, sensation-seeking and a stronger tendency to compare oneself to others [144], narcissism and psychopathy [154] have also been found to increase the likelihood of *hookups*. On the contrary, religiosity [144,155] and loneliness [156] are associated with a decreased engagement in hookups.

Regarding motivations to hookup, Garcia and Reiber [128] found that the most endorsed motivations for doing so were physical pleasure (89%), emotional gratification (54%), and to initiate a romantic relationship (51%), with no gender differences. In their sample of college students, Uecker, Pearce and Andercheck [157] reported that the most endorsed motivations were, in decreasing order, fun or excitement, sexual gratification, lack of a dating scene, hoping a relationship evolves, too busy for a relationship, and wanting to fit in. Other motivations have also been reported, such as improving one’s reputation or popularity [125]. Uecker and colleagues’ results [157] suggest that about three quarters of college students reported hooking up for a combination of sexual gratification, the lack of a dating scene, and hopes that a hookup would evolve into a romantic relationship (50%), or mainly for fun/excitement-seeking and sexual gratification only (27%). Studies further suggest that motivations vary according to gender, sex being a more common motivation for men to begin CSREs, and emotional connection a more common motivation for women [140,158].

At the event or situational level, the use of alcohol or other substances [148,159], the attractiveness of a potential hookup partner [159], and situational triggers, such as the likelihood of engaging in hookups when meeting someone at a bar or party, when someone attractive wants to hookup, or when it seems like everyone else is hooking up [127,144], have been identified as predictors of hooking up. At the environmental level, involvement in hookups has also been described as a consequence of the recent social and cultural shifts in the dating and relationship scripts, changes captured in the term “hookup culture”. The term “hookup culture” has been coined to describe the shared values, attitudes, goals, and practices surrounding *hookups* on college campuses [142,160]. Aubrey and Smith [160] have operationalized “hookup culture” through five core beliefs: (a) hooking up is harmless and best without emotional commitment; (b) hooking up is fun; (c) hooking up enhances one’s status in one’s peer group; (d) hooking up allows one to assert control over one’s sexuality; and (e) hooking up reflects one’s sexual freedom.

Comparing attitudinal and behavioral patterns between 1988–1996 and 2004–2012 waves of the U.S. General Social Survey, Monto and Carey [161] have found only modest changes that are consistent with the cultural shifts designated by “hookup culture”. Regarding the behavioral patterns, they found that respondents from the most recent survey waves did not report more sexual partners since age 18, more frequent sex, or more partners during the past year than respondents from the earlier waves. However, sexually active respondents from the more recent waves were more likely than those from the earlier waves to report sex with a casual date or a friend, and less likely to report sex with a spouse or regular partner. Regarding attitudinal patterns, Monto and Carey [161] report that recent weaves of respondents were no more accepting than earlier waves of sex between teens aged 14 to 16, sex outside of marriage, or premarital sexuality between adults; however, they were more accepting of sex between adults of the same sex. They conclude that if college students today indeed live in a so-called hookup culture, it appears to be a culture similar to the one inhabited by the earlier cohort they studied.

CSREs can describe variations in the traditional scripts toward the formation or dissolution of committed relationships. It is possible that CSRE scripts reflect different stages in the processes of dating and the development of romantic relationships, with a greater permissiveness granted for exploration before committing to such relationships. Data on motivations for hooking up suggest that only a minority of college students are completely uninterested in anything more that hooking up, and that a majority of them also report emotional motivation [128], which supports this hypothesis. However, there is a lack of follow-up data on the formation and dissolution of CSREs that would allow for such a conclusion.

*Outcomes.* The literature on CSREs mostly focuses on the *negative* outcomes of CSREs. For instance, with regards to emotions after CSREs, research show that up to three quarters of college students who have engaged in hookups report regret afterwards [159,162], with women more likely to report regret than men [162]. Such regrets notably concern the choice of partner, hopes for a relationship that did not materialize, negative social repercussions such as awkwardness or close people being hurt, guilt and moral issues, going further than expected because of alcohol, and suboptimal sexual performances [148,159,162]. Hooking up is also associated with an increase in psychological or emotional distress among young adults [156,159], possibly more among women than men [163]. CSREs have also been associated with increased risk of coercion and sexual assault [124,164], and sexually transmissible infections [159,164].

Recently, scholars have started to investigate the potentially *positive* outcomes of CSREs, given their high prevalence and popularity in emerging adulthood. An increasing body of research is now focusing on the short-term benefits of CSREs, such as feeling attractive, desirable and empowered, experiencing sexual pleasure and excitement, meeting new people, including friends and potential future romantic partners [146,165], as well as well-being, resulting in a more nuanced portrait of CSREs outcomes. For example, psychological well-being has been reported to increase among the most depressed or lonely individuals, suggesting that hooking up may be a coping strategy for internalizing problems [156]. A recent longitudinal study by Vrangalova [166] suggests that well-being following the engagement in casual sex may also depend on the motivation for doing so. Indeed, engaging in CRSEs for non-autonomous reasons was linked to lower self-esteem, higher depression and anxiety, and more physical symptoms, whereas autonomous hookup motivation (*i.e.*, emanating from one’s self) was not linked to such outcomes [166].

## 7. Sexuality in Sexual-Minority Youth

*Developmental Psychosexual Trajectories in Same-Sex Attracted Youth*. Since the 1970s and the pioneer work of Cass [167,168] and others [169,170,171], the dominant narrative of sexual identity development has suggested that sexual minority youth (SMY) progress from identity confusion to identity synthesis through various stages (*i.e.*, most often, four to six stages). Examples of such stages are: Confusion regarding sexual attraction, struggle with his/her own sexual difference, acceptance of his/her sexual orientation, sexual identity affirmation, and taking pride in oneself. Other indicators of progression in sexual identity development are the age at which they first experience specific milestones (e.g., awareness of non-exclusively heterosexual attractions, realization they might not be heterosexual, telling someone about their sexual orientation). However, stage models have been critiqued for being mostly gay male-centric, lacking empirical evidence among specific subgroups (*i.e.*, women and bisexuals), oversimplifying complex issues, not explaining the inconsistency between sexual behaviors, attractions and identity, and disregarding erotic fluidity and plasticity across the lifespan [172,173]. As a result, they tend to be abandoned in favor of models of sexual identity development that acknowledge the instability of the sexual identity over time.

In response to these critiques, Savin-Williams [173] proposed four basic tenets for a new developmental trajectory framework. First, same-sex attracted youth are similar to other adolescents in their developmental trajectories, and subject to the same biopsychosocial influences that affect youth universally. Second, same-sex-oriented youth are dissimilar from other-sex-oriented adolescents in their developmental trajectories, for both biological and cultural reasons (e.g., heteronormativity) and this dissimilarity forces them to negotiate their psychological development differently than other-sex-oriented youth. Third, same-sex-oriented youth vary in their developmental trajectories probably as much as other-sex-oriented youth when we take into account the intersectionality of their identity with gender, ethnicity, location, socioeconomic status, and so on. Fourth, each same-sex-oriented individual follows his/her own unique developmental trajectory, rendering general descriptions of group mean differences and similarities irrelevant when applied to a specific individual, and stressing the importance of adopting a person-centered approach to understanding developmental trajectories.

*Sexual Health Issues in Sexual Minority Youth (SMY)*. SMY report concerns with sexual health. Young men who have sex with men (MSM) face particularly high vulnerability to contract HIV and other STIs [174]. Condomless anal sex with HIV serodiscordant partner(s) is the main behavioral risk factor for HIV infection among MSM. Among young MSM, condomless anal sex is particularly driven by substance use, homophobia and discrimination, a lack of comprehensive sexuality education and a misconception of risks, racial and ethnic marginalization, and mental health and psychosocial issues [174,175]. While women who have sex with women (WSW), especially younger ones, are less represented in research on sexual health and sexual minorities, they nevertheless face important sexual health issues, such as unplanned pregnancies [176] and screening for the human papillomavirus (HPV) and other STIs (e.g., genital herpes and bacterial vaginosis), particularly bisexual women who also have sexual relations with men [177].

## 8. Future Directions in Youth Sexuality Research

This paper reviewed the extensive literature on sexuality in adolescence and early adulthood both *within* and *outside* romantic relationships, as well as on sexually inexperienced youth. In light of the knowledge reviewed, this paper concludes by listing four directions for future research to fill out the gaps in what is left to be known about young people’s sexuality. Firstly, historically, the literature on the development of interpersonal intimacy in adolescence and early adulthood has focused either on romantic involvement [178,179] or on sexual behavior [17]. As such, the literature on romantic relationships and youth sexuality has evolved in parallel and rather independently of each other until recently [180]. This is rather striking since sexual behaviors for most youth emerge within the context of romantic relationships [117]. As a result, much remains unknown about how characteristics of romantic relationships and partners are associated with adolescents’ and early adults’ (a)sexual behavior, despite an increasing number of scholars having raised the importance of studying youth sexuality within romantic couples [181,182,183]. As discussed in this review, a few scholars are starting to fill this gap by examining sexual behaviors in the context of romantic relationships, however this body of research is still relatively small [117,119,121,122]. More longitudinal research is needed to investigate how trajectories of romantic and sexual development run parallel to one another (e.g., timing, sequence, pace, continuity, and change), and how various stages and events in these trajectories are intertwined (for an example, see: [180]). In such future longitudinal research on youth sexuality, specific attention should be paid to assessing bidirectional relations between romance-related characteristics and processes on the one hand, and sex-related characteristics and processes on the other. This would allow for an exploration of how youth’s experiences with romantic relationships and sex are intertwined and bidirectionally influence one another over time. Further investigations of how various aspects of youth romantic relationships (and lack thereof), sexual cognitions and behaviors, romantic sex, casual sex, and asexual relationships evolve over time—including through life transitions such as, for example, puberty (e.g., [184]), schools transitions (e.g., [185,186]), entry into parenthood (e.g., [187])—would bring more complete knowledge on youth sexual development. Besides identifying main developmental trajectories, attention should be paid to the investigation of the presence or absence of sexual behaviors and experiences, both within and outside of romantic relationships, and how these may differ across subgroups of youth (e.g., boys and girls; early, middle, late adolescents and young adults; ethnicities; sexual orientation subgroups; early and late starters, adult virgins; subtypes of CSREs).

Secondly, although many studies focus on sexual intercourse, sexual behaviors encompass other types of intimate experiences as well (*i.e.*, coital and non-coital). The majority of adolescents follow a progressive sexual trajectory, where they engage in non-coital sexual behaviors before they engage in intercourse [31,32]. Thus, this narrow focus in research excludes sexually active adolescents who have not yet engaged in intercourse, but who may have engaged in other (*i.e.*, non-coital) sexual behaviors, traditionally referred to as “technical virgins” [75,77]. In addition, considering coital activities only is inherently heteronormative, and provides a limited portrait of the sexual behaviors of gay, lesbian, bisexual, and questioning youth. More in-depth life history qualitative research (for examples, see: [188,189]) focusing both on risky as well as positive sexual trajectories, and that encompass not only various behaviors (coital and non-coital, same-sex and other-sex), but also cognitions (e.g., intentions, motives), and emotions (e.g., attachment and love, desire/lust/pleasure, satisfaction, guilt, shame, regret) are needed to complement current quantitative findings. Future research should continue to acknowledge that youth sexual development is multifaceted and goes beyond the component of sexual behavior to also include cognitions and emotions. These aspects may especially advance our understanding of the reasons why adolescents have sex (e.g., intentions, motives), why they engage in risky sexual behaviors (e.g., discrepancies between the negotiation of safe sex practices and trust-scripts in romantic relationships), and how they experience sex (e.g., experienced emotions after sex), all of which remain less understood. A new body of research has started to investigate the cognitive and affective components of youth sexuality, by examining sexual decision-making and sexual agency (e.g., [190,191]), sexual intentions (e.g., [192,193], sexual emotions (e.g., [82,90], and the sexual component in one’s self-concept (e.g., [20,194], but more research in this area is needed.

Thirdly, the literature on gender, cultural, and sexual identity differences in youth sexuality reviewed in this paper reveals the importance of acknowledging between- and within-group differences and diversities, as well as similarities. One of the promising research avenues for studying the heterogeneity in youth sexual development is the monitoring of youth sexual emotions, cognitions, and behaviors with more person-centered, rather than variable-centered, methodological approaches [194,195]. Traditional methodological approaches expect samples to share their population of reference’s parameters, while person-centered methodologies examine the possible diversity and heterogeneity in the subgroups that may coexist in such samples and allow further comparisons. They have been identified as a rich complement to traditional methodological approaches [196,197]. For example, more studies in particular are needed on sexual identity development in hard-to-reach and understudied subgroups, such as questioning and gender-variant youth [198], undisclosed SMY, racialized SMY, and those who do not identify as SMY despite their same-sex behavior or attraction history, who may fall in the “mostly heterosexual” subgroup, which is now increasingly recognized to form a distinct sexual orientation group [199]. SMY are a heterogeneous group, hence the importance of adopting a person-centered approach to understanding their developmental trajectories is salient in this line of research as well (e.g., [200]). Research on SMY would also benefit from the inclusion of multidimensional questions on sexual attraction, self-identified sexual orientation and partners’ gender in general youth surveys [201]. There is also a need to increase the quality and impact of research among SMY, for instance by multiplying sources of information besides self-report data, and by implementing longitudinal designs to better understand how sexual minority identities develop over time and to identify factors, such as gender- and sexual orientation-based prejudice, that impact their sexual development and sexual health (e.g., [202]). While most research has focused on challenges to well-being, sexual health issues, the coming-out process and negative adjustment outcomes among SMY, it is paramount that we also document positive sexual development and resilience trajectories to shed light on and learn from the many SMY that successfully transition to adulthood [203,204].

Fourth, although the body of research on CSREs is vastly expanding, the extent to which CSREs reflect a cultural transformation in the way sexuality, romance, and friendship intersect nowadays, and in the norms regarding sexual agreement among sexual partners also needs to be further explored. This may for instance be done through in-depth qualitative research methods, to better theorize the impact of the macrosocial and historical factors on youth’s psychosexual development (e.g., [136,205]). Another avenue that would not only reduce memory biases and retrospective reconstructions of events based on more recent experiences [206] but also enable the examination of the convergence in partners’ reports while also tapping the subjectivity of each member’s experiences, would be to collect data on CSREs with both sexual partners shortly after they occurred, for example by using digital diary apps or beepers at random times with both partners to assess their thoughts and perceptions on their sexual partnership. Such methods are already being used to investigate individual experiences of CSREs (e.g., [207,208]. In general, more dyadic (*i.e.*, romantic and sexual partners) research is needed to assess how each individual of a romantic or sexual dyad initiates and develops sexual relations over time, including partner selection processes, socialization, and use of modern technological tools designed for mating and sexual encounters, such as dating sites and apps, which may be especially relevant for SMY (e.g., [209,210]). Besides multi-informant, macro-time (e.g., questionnaires) and micro-time (e.g., daily diaries) longitudinal data on individual, partner, *and* dyadic couple characteristics and processes (and their interrelated changes over time), observational research methods could provide valuable additional data on micro-interactions between romantic or sexual partners (e.g., communication, negotiation, support, conflict resolution, level of agreement towards their sexual status). All in all, important next steps in research on sexuality in adolescence and young adulthood will encompass a rich combination of multimethod and multi-informant research methods.

## 9. Conclusions

Overall, the reviewed body of literature has provided important empirical knowledge and theoretical understanding on adolescent and early adult sexuality. This knowledge is valuable for the design of effective and empirically-based comprehensive sexuality education programs, and suggests ways in which parents, teachers, sex educators and practitioners in adolescent medicine can support adolescents’ healthy, responsible and positive exploration of their sexuality, and promote relational and sexual health (e.g., [24,25,26]). It also sheds light on the importance of acknowledging the many possible forms in which adolescents and young adults experience and experiment (or do not experiment) with intimacy and sexuality, and on the multitude of social norms that present-day adolescents and young adults learn to navigate in their romantic relationships and sexual development.

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
