# Peer review of "Sexuality (and Lack Thereof) in Adolescence and Early Adulthood: A Review of the Literature"

_behavsci, 2016, doi:10.3390/bs6010008_

Round 1

Reviewer 1 Report

See attached

Reviewer 2 Report

This paper addresses a very important issue in the field of relationships research, namely the overlap between sexuality and romantic relationships during the adolescent and young adult periods. The authors correctly point out that two fields of research currently exist, one focused on sexuality and the other on romantic relationships. Since sexuality is an inherent feature of romance, an important direction for future research is to integrate these two fields. This paper has the potential of guiding further research in this area.

            The strengths of this paper include a very extensive and comprehensive review of current research on adolescent/young adult sexuality. The research covers a large array of topics including risky and normative sex, timing, early and late onset, asexuality, gender.  In addition sexuality is considered within romantic relationships as well as non-romantic sexuality.

            Despite these strengths, the paper in its present form, does not fully live up to its potential as a bridge in the romance and sexuality fields.  This is because the paper is largely focused on sexuality and only very briefly focused on the relevant research on romantic relationships. Indeed of the 20 pages of text, roughly 2 refer to romantic relationships and 2 refer to sexuality within romantic relationships. I would like to suggest that a re-organization of the chapter might be helpful. Specifically, the topics which comprise the review of sexuality research could used as an overall structure within which to discuss issues pertaining to romantic relationships as well as sexuality.  For example, there is research on duration, onset, timing, quality, abstinence etc within each field. Reviewing the research for each field within these topics would allow the authors to highlight the points of intersection as well as the points of departure. Indeed this kind of organization would be in line with the authors’ recommendations for future research, noted on page 19.

            The research on CSRES (casual sexual relationships and experiences) which is reviewed in the paper is interesting. However the relevance of this research to the authors’ stated goals of bridging the fields of sexuality and romance is not clear.  Unless this material can be integrated into the overall intent of the paper it would probably be best to eliminate this section as it distracts from the paper’s overall intent.

            It is good to see the authors address sexual minority youth. However the implications of this research for the paper topic are not clear. Perhaps following a similar framework as above would help t make this point clearer.  As well, it is not clear why sexual minority youth are selected for inclusion while other non-mainstream experiences are not included. In particular the sexual and romantic experiences of cultural minorities and youth from other cultures would be important to consider.

Round 2

Reviewer 1 Report

I appreciate the thoroughness of the revisions and responses to my comments. I think the focus of the paper is improved by reviewing sexuality in adolescence with some attention to its interaction with romantic relationships. I believe this is a good review of the literature on this topic and the authors make important points about future steps in this field. 

Reviewer 2 Report

I have reviewed the author response to the revisions and I ahve quickly read the revised paper. I think this reorganization responds very well to the concerns that I raised previously.  An excellent paper